

# Characterization of the transcriptome and EST-SSR development in *Boea clarkeana*, a desiccation-tolerant plant endemic to China

Ying Wang[1,2], Kun Liu[1,2], De Bi[1], Shoubiao Zhou[1,3] and Jianwen Shao[1,2]

[1] College of Life Sciences, Anhui Normal University, Wuhu, Anhui, China
[2] Anhui Provincial Key Laboratory of the Conservation and Exploitation of Biological Resources, Anhui Normal University, Wuhu, Anhui, China
[3] College of Environmental Science and Engineering, Anhui Normal University, Wuhu, Anhui, China

## ABSTRACT

**Background**. Desiccation-tolerant (DT) plants can recover full metabolic competence upon rehydration after losing most of their cellular water (>95%) for extended periods of time. Functional genomic approaches such as transcriptome sequencing can help us understand how DT plants survive and respond to dehydration, which has great significance for plant biology and improving the drought tolerance of crops. *Boea clarkeana* Hemsl. (Gesneriaceae) is a DT dicotyledonous herb. Its genomic sequences characteristics remain unknown. Based on transcriptomic analyses, polymorphic EST-SSR (simple sequence repeats in expressed sequence tags) molecular primers can be designed, which will greatly facilitate further investigations of the population genetics and demographic histories of DT plants.

**Methods**. In the present study, we used the platform Illumina HiSeq™2000 and *de novo* assembly technology to obtain leaf transcriptomes of *B. clarkeana* and conducted a BLASTX alignment of the sequencing data and protein databases for sequence classification and annotation. Then, based on the sequence information, the EST-SSR markers were developed, and the functional annotation of ESTs containing polymorphic SSRs were obtained through BLASTX.

**Results**. A total of 91,449 unigenes were generated from the leaf cDNA library of *B. clarkeana*. Based on a sequence similarity search with a known protein database, 72,087 unigenes were annotated. Among the annotated unigenes, a total of 71,170 unigenes showed significant similarity to the known proteins of 463 popular model species in the Nr database, and 59,962 unigenes and 32,336 unigenes were assigned to Gene Ontology (GO) classifications and Cluster of Orthologous Groups (COG), respectively. In addition, 44,924 unigenes were mapped in 128 KEGG pathways. Furthermore, a total of 7,610 unigenes with 8,563 microsatellites were found. Seventy-four primer pairs were selected from 436 primer pairs designed for polymorphism validation. SSRs with higher polymorphism rates were concentrated on dinucleotides, pentanucleotides and hexanucleotides. Finally, 17 pairs with stable, highly polymorphic loci were selected for polymorphism screening. There was a total of 65 alleles, with 2–6 alleles at each locus. Primarily due to the unique biological characteristics of plants, the $H_E$ (0–0.196), $H_O$ (0.082–0.14) and PIC (0–0.155) per locus were very low. The functional annotation distribution centered on ESTs containing di- and tri-nucleotide SSRs,

Corresponding authors
Shoubiao Zhou,
zhoushoubiao@vip.163.com
Jianwen Shao, 545491044@qq.com

and the ESTs containing primers BC2, BC4 and BC12 were annotated to vegetative dehydration/desiccation pathways.

**Discussion**. This work is the first genetic study of *B. clarkeana* as a new plant resource of DT genes. A substantial number of transcriptome sequences were generated in this study. These sequences are valuable resources for gene annotation and discovery as well as molecular marker development. These sequences could also provide a valuable basis for future molecular studies of *B. clarkeana*.

# INTRODUCTION

Resurrection plants are desiccation-tolerant (DT), which enables them to recover full metabolic competence upon rehydration after losing most of their cellular water (>95%) for extended periods of time (*Farrant, Brandt & Lindsey, 2007*). Though non-vascular plants and spores of tracheophytes are commonly DT (*Rodriguez et al., 2010*), this feature is rare in angiosperms (*Gaff, 1971*; *Porembski & Barthlott, 2000*; *Proctor & Pence, 2002*). The mechanisms of DT are different between the extant lower orders and angiosperms (*Farrant, Brandt & Lindsey, 2007*). Understanding how DT plants survive and respond to dehydration has great significance for plant biology and crop drought tolerance improvement, which could contribute to future water resource management decisions. Moreover, research on DT angiosperms could inform crop cultivation (*Farrant, Brandt & Lindsey, 2007*; *Oliver et al., 2011a*; *Gechev et al., 2012*; *Xiao et al., 2015*). In recent decades, research has focused on revealing the physiological and molecular mechanisms of DT in angiosperm plants and their recovery processes (*Bianchi et al., 1993*; *Bernacchia, Salamini & Bartels, 1996*; *Sherwin & Farrant, 1998*; *Cooper & Farrant, 2002*; *Collett et al., 2003*; *Collett et al., 2004*; *Schneider et al., 2003*; *Alcazar et al., 2011*; *Oliver et al., 2011a*; *Oliver et al., 2011b*; *Christ et al., 2014*; *Zhu et al., 2015*). While a functional genomic approach, such as transcriptome sequencing, could be fruitful for exploring the mechanisms of DT (*Xiao et al., 2015*), transcriptomics could identify the metabolic processes involved in DT. Gene Ontology (GO, http://www.blast2go.com/b2ghome) and Cluster of Orthologous Groups (COG, http://www.ncbi.nlm.nih.gov/COG/) analyses can also help us understand the distribution of functional genes in plants at the macro level (*Conesa et al., 2005*; *Ye et al., 2006*). Moreover, the gene products of metabolic processes and the functions of genes related to cellular processes can be detected by BLASTX using the KEGG database (Kyoto Encyclopedia of Genes and Genomes, http://www.kegg.jp/) (*Kanehisa et al., 2008*). These approaches can help us study gene behaviors in biologically complex processes, such as vegetative dehydration/desiccation pathways, in DT plants (*Xiao et al., 2015*). However, to the best of our knowledge, only a few gene expression and EST sequencing studies have been performed in angiosperms with DT, including the dicot species *Craterostigma plantagineum* (*Bockel, Salamini & Bartels, 1998*; *Rodriguez et al., 2010*),

*Boea hygrometrica* (*Xiao et al., 2015*; *Zhu et al., 2015*), and *Haberlea rhodopensis* (*Gechev et al., 2013*) and the monocot species *Sporobolus stapfianus* (*Neale et al., 2000*; *Le et al., 2007*), *Xerophyta viscosa* (*Mundree et al., 2000*; *Mowla et al., 2002*; *Lehner et al., 2008*), and *X. humilis* (*Collett et al., 2004*; *Illing et al., 2005*; *Mulako et al., 2008*).

*Boea* (Gesneriaceae) is a rare group of resurrection plants within angiosperms (*Liu, Hu & Zhao, 2007*; *Xiao et al., 2015*). *B. clarkeana* Hemsl. is a desiccation-tolerant herb endemic to China. The whole plant, detached leaf and leaf segment all retain the DT phenotype, and the excellent drought tolerance of this plant has been of concern in the last few years (*Chao et al., 2013*; *Zhang et al., 2016*). *B. clarkeana* is a small perennial dicotyledonous plant that is mainly distributed in eight provinces and one municipality along the middle-lower reaches of the Yangtze River in China (*Li, 1996*; *Li & Wang, 2005*). It is found only on rock outcrops (such as inselbergs) among some lithophytes, where dehydration occurs frequently (*Jenks & Wood, 2007*). It is commonly used as a medicinal plant to treat traumatic hemorrhage and traumatic injury (*Li & Wang, 2005*). However, genomic sequences of *B. clarkeana* are scarce, and only a few nucleotide sequences are found in public databases (http://www.ncbi.nlm.nih.gov/). To fill this critical gap and obtain the first genomic resources, we used the Illumina HiSeq$^{TM}$2000 platform and *de novo* assembly to obtain leaf transcriptomes of *B. clarkeana* and conducted a BLASTX (http://blast.ncbi.nlm.nih.gov/Blast.cgi) alignment of the sequencing data and protein databases for sequence classification and annotation.

We also assessed the SSRs, or microsatellites, that are distributed universally in gene coding and non-coding regions. As the major tool of genetic research, the neutral SSR markers are widely used in botanic sciences and functional SSR markers might affect gene function by influencing recombination and chromatin organization, regulating gene expression and activity, etc. (*Cavagnaro et al., 2010*; *Li et al., 2012a*; *Zalapa et al., 2012*). In DT plants, identifying functional genes that contain functional SSRs could help us to understand the evolution and expression of these genes, making SSRs a valuable resource for genetic studies (*Li et al., 2002*; *Gupta et al., 2003*). Therefore, based on the *B. clarkeana* transcriptome, 17 pairs of polymorphic EST-SSR molecular primers were developed and characterized. The results of this study will greatly facilitate further investigations of the genetics and demographic histories of populations of this DT plant.

## MATERIALS AND METHODS

### Plant materials and genomic DNA extraction

The materials of 11 natural populations were sampled from six provinces and one municipality in China that covered the vast majority of the natural habitats of *B. clarkeana* (*Li & Wang, 2005*). Young leaves were collected, rapidly dried and preserved in silica gel. DNA extraction was carried out using the QIAGEN® DNeasy® Plant Mini Kit (QIAGEN, Germany).

### RNA isolation and cDNA library construction

The young leaves of three individual *B. clarkeana* plants from the population of Mt. Fenghuang in Anhui Province (30°88′N, 118°02′E) were collected, mixed and frozen in

liquid nitrogen; then, the sampled tissues were stored at $-80\,°C$ until RNA extraction. Total RNA was isolated using a TRIzolKit (Life Technologies, USA) and DNase I (TaKaRa, Japan) following the manufacturer's protocols. After total RNA was obtained, mRNA + poly (A) were isolated using beads with Oligo (dT), and fragmentation buffer was added to cut the mRNA into short fragments. Then, first-strand cDNA was obtained from the RNA sequence fragments using reverse transcriptase and random primers (Invitrogen, Carlsbad, CA), and second-strand cDNA was synthesized using buffer, dNTPs, RNaseH and DNA polymerase I. Following the ligation of adapters, a single 'A' base was added to the 3' end of these cDNA fragments to facilitate end repair. Based on the amplification of these products, the cDNAs were separated on an agarose gel, and the cDNA library was generated.

## Sequencing and *de novo* assembly

The raw reads were produced from the cDNA library using an Illumina HiSeq[TM]2000 genomic sequencer at the Beijing Genomics Institute (BGI, Shenzhen, China, http://www.genomics.cn/index). The subsequent analysis was based on the clean reads generated by filtering the raw reads. We used the filter_fq program (BGI, Shenzhen, China) to remove reads with more than 5% unknown nucleotides (N) and low-quality sequences with more than 20% low-quality bases (quality value $\leq10$) and adaptors to obtain clean reads. Then, we used the short read assembly program Trinity (Release-2013-02-25, http://trinityrnaseq.sourceforge.net/) for *de novo* transcriptome assembly by combining the clean reads into contigs with a sequence fragment length of 200 bp ($\pm25$ bp) (*Grabherr et al., 2011*). Two contigs were then connected into a single scaffold, and we called the resulting sequences unigenes. These unigenes were removed to prevent redundancy with TGICL (version 2.1) and further spliced to generate non-redundant unigenes that were as long as possible (*Pertea et al., 2003*). The raw sequencing data with accession number SRX1600046 were deposited in the Sequence Read Archive (SRA) of National Center for Biotechnology Information (NCBI), which will be released upon publication.

## Functional annotation and classification of unigenes

BLASTX alignment ($E$-value $< 10^{-5}$) between the unigenes and protein databases, such as NCBI non-redundant protein (Nr), GO, COG, and KEGG, was performed to annotate and classify the transcriptome. Based on the Nr database annotation, we used Blast2GO v2.5.0 to obtain GO terms with an $E$-value threshold of $10^{-5}$ (*Conesa et al., 2005*). With the Web Gene Ontology Annotation Plot (WEGO) (*Ye et al., 2006*), the distributions of GO terms were plotted to describe the categories, and the unigenes were aligned to the COG database for possible functional prediction and classification. The unigenes containing SSRs were also aligned to euKaryotic Orthologous Groups (KOGs) via BLASTX. Finally, we mapped the unigenes to each level 3 pathway graph using the KEGG database to obtain pathway annotations for the unigenes.

## EST-SSR mining, primer design and polymorphism identification

SSRs from unigenes were detected and located using MIcroSAtellite (MISA, http://pgrc.ipk-gatersleben.de/misa/misa.html) (*Zalapa et al., 2012*). Compound SSRs (two or more SSRs

in which the interval was no more than 100 bp) were excluded, and only SSRs with flanking sequences longer than 150 bp and containing two to six repeat motifs were considered. The mono-, di-, tri-, tetra-, penta- and hexa-nucleotide motif SSRs with a minimum of twelve, six, five, five, four and four repeats, respectively, were detected. We designed primer pairs using the online program Primer3.0 (http://www.onlinedown.net/soft/51549.htm) with the following criteria: (1) a product sequence length of 100–300 bp and no secondary structure; (2) a primer length of 18–28 bp with an optimum length of 23 bp; (3) a Tm of 55–65 °C with an optimum Tm of 60 °C and a difference between the Tm values of the forward and reverse primers of no greater than 4 °C; and (4) a guanine-cytosine (GC) content of 40–60%, with 50% as the optimum. For other parameters, the default settings were used.

Seventy-four primer pairs divided into two groups were selected for DNA amplification. The first group of 50 primer pairs was randomly selected for amplification, and the motifs that had more polymorphic alleles in the first group were used to increase the selected ratio in the second 24 primer pairs. The mixed DNA from three individuals from different populations of *B. clarkeana* was used to verify the amplification products, and the primers that amplified successfully were chosen for primary polymorphism identification. Using these primers, amplification was conducted using 12 individuals from 11 natural populations. Then, the DNA of 128 individuals from 11 populations were amplified using primer pairs that had more polymorphic loci for further identification of polymorphisms. The ESTs containing SSRs were aligned to GO, COG, and KEGG databases through BLASTX to help us understand the functional annotations of the sequences.

We used fluorescently-labeled (6-FAM, HEX, TAMRA or ROX) M13-tailed (5'-TGTAAAACGACGGCCAGT-3') primers to accurately screen the variation among individuals. PCR was performed in a 15-μL reaction containing 2.5 mM MgCl$_2$ and dNTP (TaKaRa, Dalian, China), 0.5 U of *Taq* polymerase (TaKaRa, Dalian, China), 1 × PCR buffer, and 50 ng of genomic DNA. The primers included 0.04 μM forward primers, 0.01 μM M13-tailed reverse primers, and 0.04 μM M13 primers with fluorescent tails. The annealing temperature was different for each locus. We used 54 °C as the unified annealing temperature for PCR, and the amplification conditions were as follows: initial denaturation at 94 °C for 5 min; 35 cycles of 30 s at 94 °C, 40 s annealing at 54 °C, and 45 s elongation at 72 °C; and a final extension at 72 °C for 10 min. After screening on a 1.0% agarose gel, the sequence typing of successfully amplified products was performed using an ABI 3730 DNA Analyzer (Applied Biosystems, Foster City, California, USA). Then, we manually scored alleles using GeneMarker software (version 2.2.0).

Deviations from Hardy-Weinberg equilibrium (HWE) were calculated using the online tool GENEPOP (http://www.genepop.curtin.edu.au/) with Bonferroni's correction. The number of alleles ($N_A$) was calculated using MicroChecker (version 2.2.3). The expected ($H_E$) and observed heterozygosity ($H_O$) of each locus were detected by GenALEx 6 (*Peakall & Smouse, 2006*), and the polymorphism information content (PIC) was calculated using the PowerMarker program (version 3.25) (*Liu & Muse, 2005*). Then, neutral markers were detected using LOSITAN (*Beaumont & Nichols, 1996*; *Antao et al., 2008*).

**Table 1  Summary of sequence assembly using Illumina sequencing.**

| Sequence | Items | Value |
|---|---|---|
| **Reads** | Total raw reads | 110,834,050 |
| | Total clean reads | 104,021,494 |
| | Total clean nucleotides (nt) | 9,361,934,460 |
| | Q20 percentage (%) | 97.55 |
| | N percentage (%) | 0 |
| | GC percentage (%) | 45.43 |
| **Contig** | Total number | 94,546 |
| | Total length (nt) | 46,012,409 |
| | Mean length (nt) | 487 |
| | Contig N50 (nt) | 1,075 |
| **Unigene** | Total number | 91,449 |
| | Total length (nt) | 148,176,175 |
| | Mean length (nt) | 1,620 |
| | Unigene N50 (nt) | 2,389 |
| | Distinct clusters | 55,888 |
| | Distinct singletons | 35,561 |

# RESULTS

## Illumina sequencing and *de novo* assembly

Sequencing success was determined by the length of the reads, as longer reads would increase the probability of SSRs being discovered (*Zalapa et al., 2012*). A total of 9,361,934,460 nt bases were generated in this study. After cleaning and quality checks, we obtained 104,021,494 clean reads with Q20 bases (sequences with sequencing error rates <1%) at 97.55%, and the N (ambiguous bases) and GC contents were 0 and 45.43%, respectively. *De novo* assembly was performed using the Trinity program; a total of 94,546 contigs were generated with an average length of 487 nt and an N50 value of 1,075 nt. Finally, a total of 91,449 unigenes with a total length of 148,176,175 nt were detected, and the average length and N50 were 1,620 nt and 2,389 nt, respectively. The final assembled transcripts of *B. clarkeana* were longer than those of its sibling species, i.e., the *Primulina* species and *B. hygrometrica*, which were assembled using Illumina (*Ai et al., 2015*) and the 454 pyrosequencing platform (*Zhu et al., 2015*), respectively. As 454 pyrosequencing produces longer reads than Illumina, the sequencing results were ideal in this study (*Zalapa et al., 2012*). A summary of the sequence assembly after Illumina sequencing is shown in Table 1. The sequence-length distribution of the unigenes is shown in Fig. 1.

## Functional annotation and classification of unigenes

For function annotation analysis, we obtained 71,170, 59,962, 32,336 and 44,929 unigenes annotated to the Nr, GO, COG and KEGG databases, respectively. In total, 72,078 unigenes (78.82% of all assembled unigenes) were successfully annotated in the present study. This number of successful annotations was more than those reported for other DT plants, including *B. hygrometrica* (66.6% (*Zhu et al., 2015*) and 47.09% (*Xiao et al., 2015*)) and

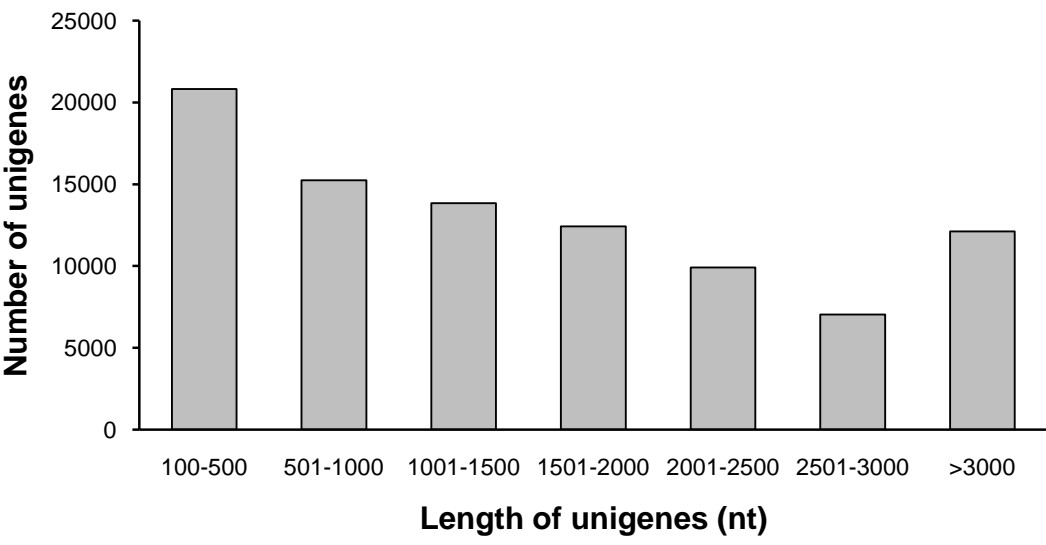

**Figure 1** The length distribution of unigenes.

*Syntrichia caninervis* (58.7%) (*Gao et al., 2014*), which indicates that the functions of genes in *B. clarkeana* are better conserved in this study.

## Nr annotation

In total, 71,170 unigenes were annotated from 463 popular model species in the Nr databases. The species distribution of Nr annotations (Fig. 2) were mainly comprised of *Lycopersicon esculentum* (35.11%), *Vitis vinifera* (27.81%), *Amygdalus persica* (6.66%), castor bean (*Ricinus communis*; 6.12%), black cottonwood (*Populus balsamifera* subsp. *trichocarpa*; 5.2%), *Fragaria vesca* subsp. *vesca* (3.18%) and *Glycine max* (2.82%). The most common species found in terms of this similarity were those of Solanaceae, with which six species had similar genes (26,585, 37.35%). Only a small fraction of all transcripts showed similarities to genes in other species. The structural features of the protein-coding genes were similar to those of their homologs in other previously studied DT plants, including *C. plantagineum* (*Rodriguez et al., 2010*), *B. hygrometrica* (*Zhu et al., 2015*) and *H. rhodopensis* (*Gechev et al., 2013*). The species distribution of Nr annotations primarily consisted of *V. vinifera*, *R. communis* and *P. trichocarpa*, which showed significant homology; however, *B. clarkeana* showed some differences in our study, which indicated that *B. clarkeana* shares a common origin with *L. esculentum* and *V. vinifera*.

## GO and KEGG classification

Based on the Nr annotations, 59,962 unigenes (65.57% of all unigenes) were assigned to three ontologies and subdivided into 55 functional GO terms. The annotation scale in *B. clarkeana* was much greater than that in the related species *B. hygrometrica* (28.71% (*Xiao et al., 2015*); and 43.7% (*Zhu et al., 2015*)). Similarly to previous studies, the 'Biological process' (49.45%) was the main ontology, followed by 'Cellular component' and 'Molecular function' ontologies (37.11% and 13.43%). A high percentage of genes were classified under the GO terms 'Cellular process', 'Metabolic process', 'Cell', 'Cell part', 'Organelle',

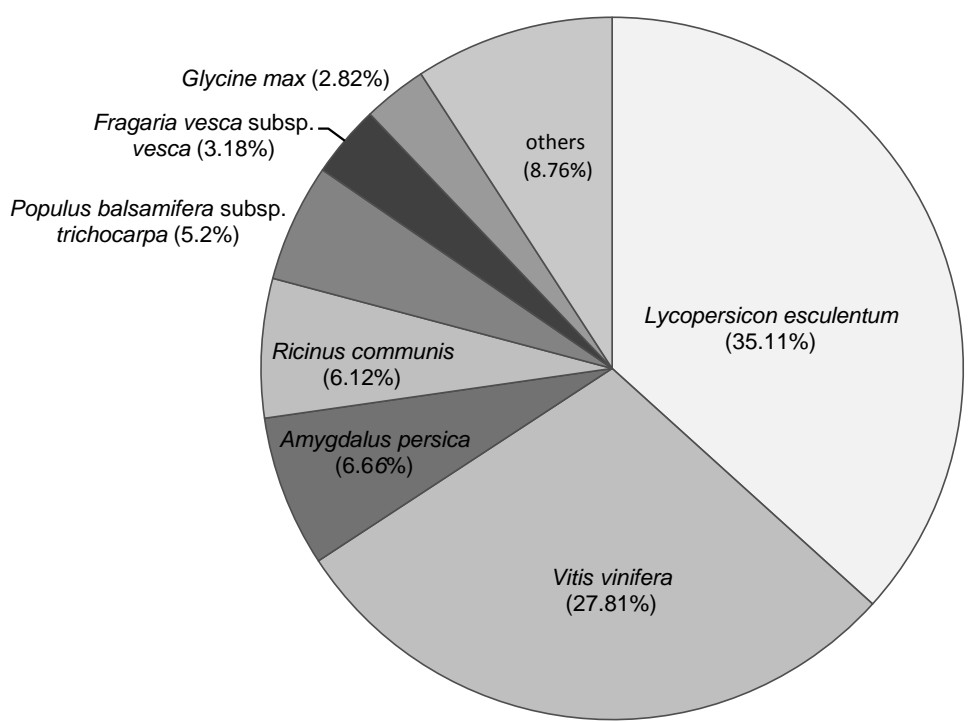

**Figure 2** **The species distribution of Nr annotations.**

'Catalytic activity' and 'Binding' (*Gupta et al., 2003*; *Durand et al., 2010*; *Blanca et al., 2011*; *Li et al., 2012a*; *Xiao et al., 2015*; *Zhu et al., 2015*). The assignment of GO terms in *B. clarkeana* in this study focused on 'Single-organism process', 'Physiological response to stimulus', 'Biological regulation', 'Localization', 'Macromolecular complex', 'Symplast' and 'Transporter activity', which reflected the functional gene expression characteristics of *B. clarkeana* during normal growth. Compared with the related species *B. hygrometrica* under different DT treatments, there were more functional GO terms, more dispersed gene distributions, and different sets of GO terms, especially in the 'Molecular function' ontology (Fig. 3) (*Xiao et al., 2015*; *Zhu et al., 2015*). This result was mainly due to selective gene expression caused by the adaptation of cells to various physiological states and environmental changes.

Based on sequence homology searches against the KEGG database, 44,924 unigenes (49.12% of all unigenes) were mapped in 128 pathways. The enrichment of the KEGG annotation in this study was much greater than that of *B. hygrometrica* (24.43% (*Xiao et al., 2015*); 15.1% (*Zhu et al., 2015*)). Among these pathways, 'Metabolic pathway' (9,232, 20.55% of KEGG unigenes) and 'Metabolic biosynthesis of secondary metabolites' (3,764, 8.38%) were the largest categories of 'Metabolism'. The greatest highlight of the KEGG analysis in our study was the enrichment of the following vegetative dehydration/desiccation pathways: 'Plant-pathogen interaction' (1,769 unigenes, 3.94% of KEGG unigenes) in the pathogen defense system; 'Glycerophospholipid metabolism' (803, 1.79%) in vesicular trafficking for protein receptor interactions; 'Plant hormone signal transduction' (1,783, 3.97%) for abiotic stress responses; the mRNA surveillance (1,027, 2.29%) pathway for
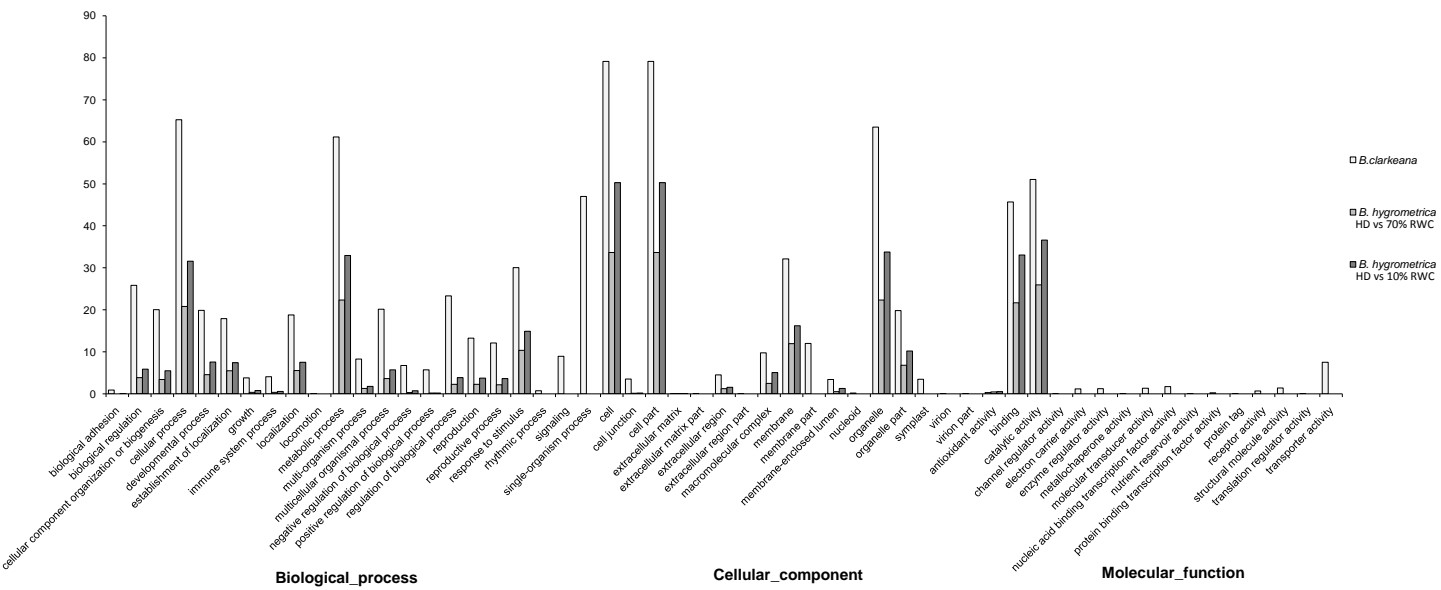

**Figure 3** **Gene ontology classification of unigenes between _B. hygrometrica_ and _B. clarkeana_.** GO functions are shown in the _X_-axis. The _Y_-axis shows the percentage of genes annotated with the GO function. RWC, relative water content; HD, hydrated. The data of _B. hygrometrica_ came from _Xiao et al. (2015)_.

damaged transcript removal; 'Photosynthesis' (154, 0.34%) and 'Nitrogen metabolism' (154, 0.34%) for the depletion of transcripts during dehydration. In addition, some other environment-related pathways, including 'Phosphatidylinositol signaling system' (535, 1.19%), 'ABC transporters' (499, 1.11%) and 'Circadian rhythm-plant' (377, 0.84%) were also enriched. These results indicate that in normal metabolic processes, _B. clarkeana_ maintains its abundant vegetative dehydration/desiccation pathways. The results of our study are consistent with those of other studies, which identified the plant genes and gene products with central roles in DT (_Gechev et al., 2012_; _Xiao et al., 2015_).

## COG and KOG classification of unigenes with SSRs

In total, 56,493 functionally annotated unigenes from 32,336 (35.36% of all unigenes) COG unigenes were assigned to 25 possible functional categories in COG annotations (Fig. 4). Among the categories, the largest group was the cluster for 'General function prediction only' (10,438, 32.28%), followed by 'Replication, recombination and repair' (5,561, 17.20%) and 'Transcription' (5,322, 13.46%). The smallest groups were 'Cell motility' (228, 0.71%), 'Extracellular structures' (17, 0.05%) and 'Nuclear structure' (14, 0.04%). This pattern is similar for some angiosperms, including _Camelina sativa_ (_Liang et al., 2013_), _Apium graveolens_ (_Fu, Wang & Shen, 2013_) and _Chrysanthemum nankingense_ (_Wang et al., 2013_). The 'Replication, recombination and repair' (17.20%) category has abundant genes in _B. clarkeana_, and this plant showed more repaired genes.

After SSR detection using the MIcroSAtellite (MISA) software with unigenes as references, 7,610 unigenes carrying 8,563 SSRs were found. Then, 3,267 unigenes with SSRs had hits in 24 categories of the KOG database without 'Nuclear structure'. Among 24 categories, the largest group was 'General function prediction' (1,166, 35.69% of
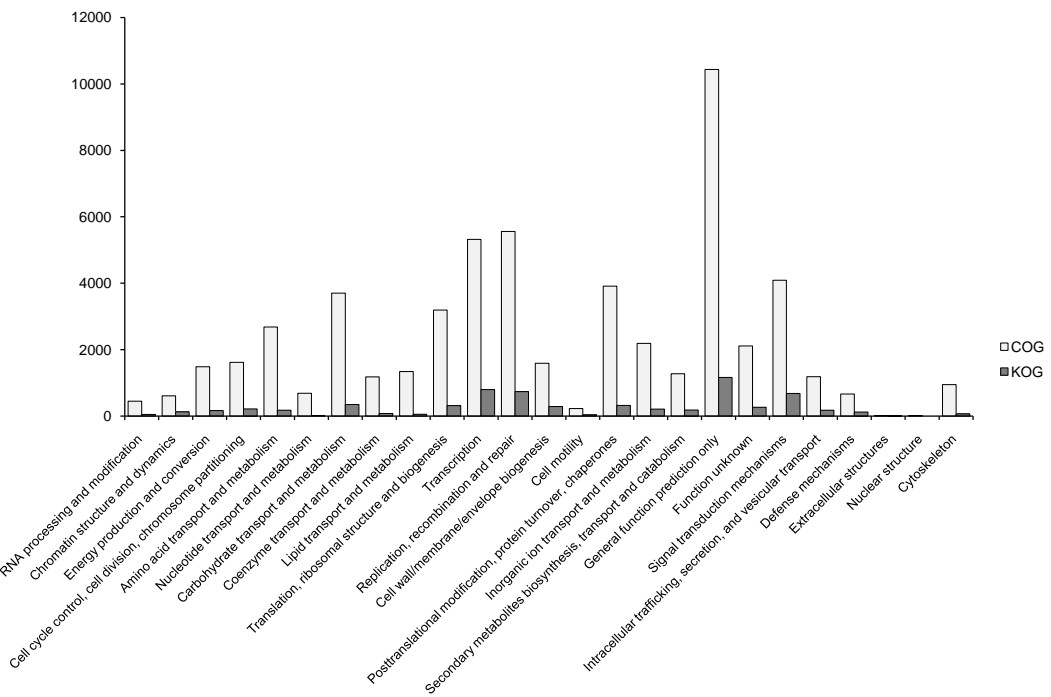

**Figure 4  COG and KOG functional classification of unigenes.** The horizontal coordinates are functional classes of COG and KOG, and the vertical coordinates are numbers of unigenes in one class.

unigenes with SSRs in KOG), followed by 'Transcription' (797, 24.40%), 'Replication, recombination and repair' (737, 22.56%) and 'Signal transduction mechanisms' (684, 20.94%). Compared with other studies of EST-SSRs (*Li et al., 2012a*; *Liang et al., 2013*; *Liu et al., 2013*), 'Replication, recombination and repair' and 'Signal transduction mechanisms' (684, 20.94%) were highlighted in *B. clarkeana*. These 3,267 ESTs will provide a valuable repository of abundant information for future functional SSR studies.

## Frequency and distribution of SSRs

All 91,449 assembled unigenes were used to mine potential SSRs in this study, and a total of 7,610 unigenes containing 8,563 SSRs were identified. Other reports have identified approximately 2,000 EST-SSRs using NGS (Next-Generation Sequencing) (*Liu et al., 2013*; *Wang et al., 2013*; *Xiang et al., 2015*); the quantity of EST-SSRs in our study was significantly larger, which was probably due to the use of longer reads and the expression characteristics of the species (*Zalapa et al., 2012*).

Among those unigenes containing SSRs, 338 SSRs presented a compound formation, and 812 unigenes contained more than one SSR. On average, one SSR was found every 17.30 kb. Among the identified SSRs, dinucleotide motifs were the most abundant (3,991, 46.61% of all SSRs), followed by mono- (2,163, 25.26%), tri- (1,957, 22.85%), hexa- (267, 3.12%), tetra- (198, 2.3%), and penta- (36, 0.42%) nucleotide motifs. This result was similar to the findings reported for *A. graveolens* (*Fu, Wang & Shen, 2013*) and *Hevea brasiliensis* (*Li et al., 2012a*). The distributions and frequencies of different motifs are shown in Fig. 5.

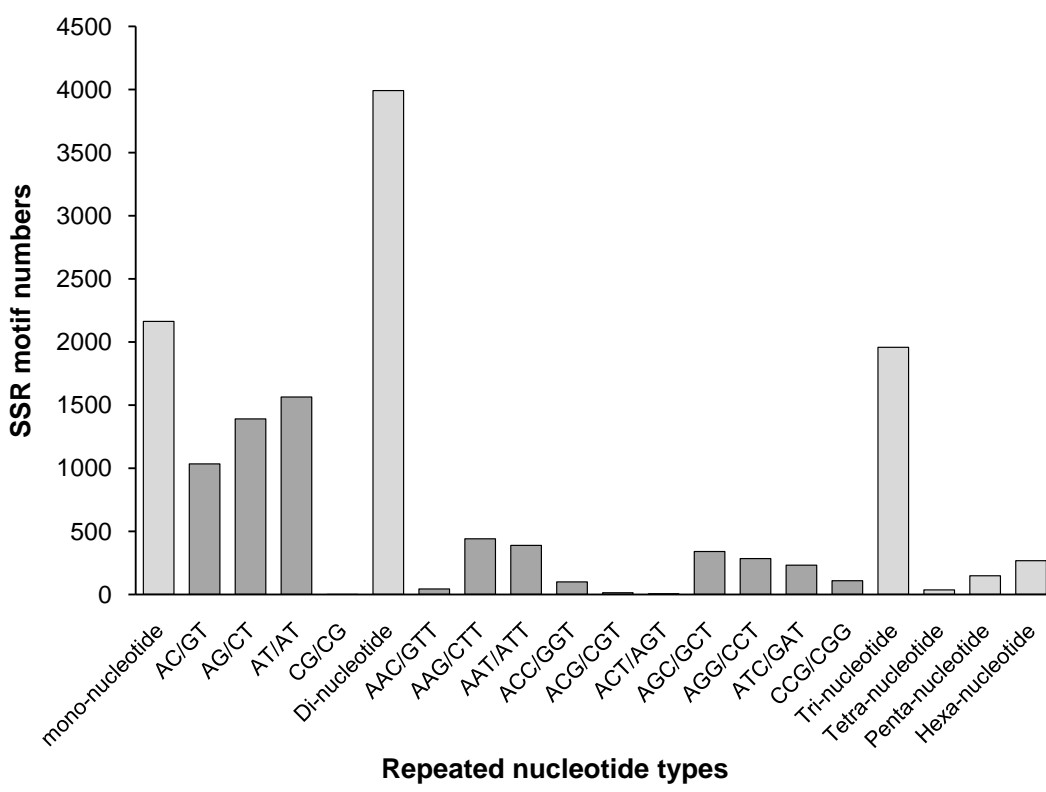

**Figure 5** The distribution of the most repeated nucleotide types.

Among all SSR loci, 109 different motifs were identified. The largest subset of mononucleotides were A/T (2,093, 24.44% of all SSRs), and there were only 70 C/G nucleotides in total. Of the dinucleotides, AT/AT (1,564, 18.26%) and AG/CT (1,391, 16.24%) were roughly equivalent, followed by AC/GT (1,035, 12.09%). Of the trinucleotides, AAG/CTT (441, 5.15%) was the most common, followed by AAT/ATT (389, 4.54%) and AGC/GCT (341, 3.98%). The ACAT/ATGT (18, 0.21%) motif comprised the most common tetranucleotide, and the most common pentanucleotides and hexanucleotides were AAAAG/CTTTT (42, 0.49%) and AAGAGC/GCTCTT (68, 0.79%), respectively. The repeat numbers of most SSRs ranged from four to twelve, and the most frequent repeat number was six (2,066, 24.13%), followed by five (1,233, 14.40%) and seven (1,113, 13.00%). Furthermore, the length of SSRs ranged from 12 to 25 bp (Fig. 6). Among the di- and trinucleotides, the most common lengths were 12 bp and 15 bp, respectively. The longest di-, tri- or tetranucleotide was 24 bp, whereas the longest pentanucleotide was 25 bp in length; all hexanucleotides were 24 bp.

## Development and validation of polymorphic SSR markers

As a result, a total of 436 (only 5.73% of SSR-containing ESTs) eligible primer pairs (mononucleotide, 1; di-, 191; tri-, 205; tetra-, 5; penta-, 12; hexa-, 22) were designed. Primers could not be successfully designed for the other sequences, primarily due to their overly long sequence lengths and insufficient flank lengths, as well as the abundance of
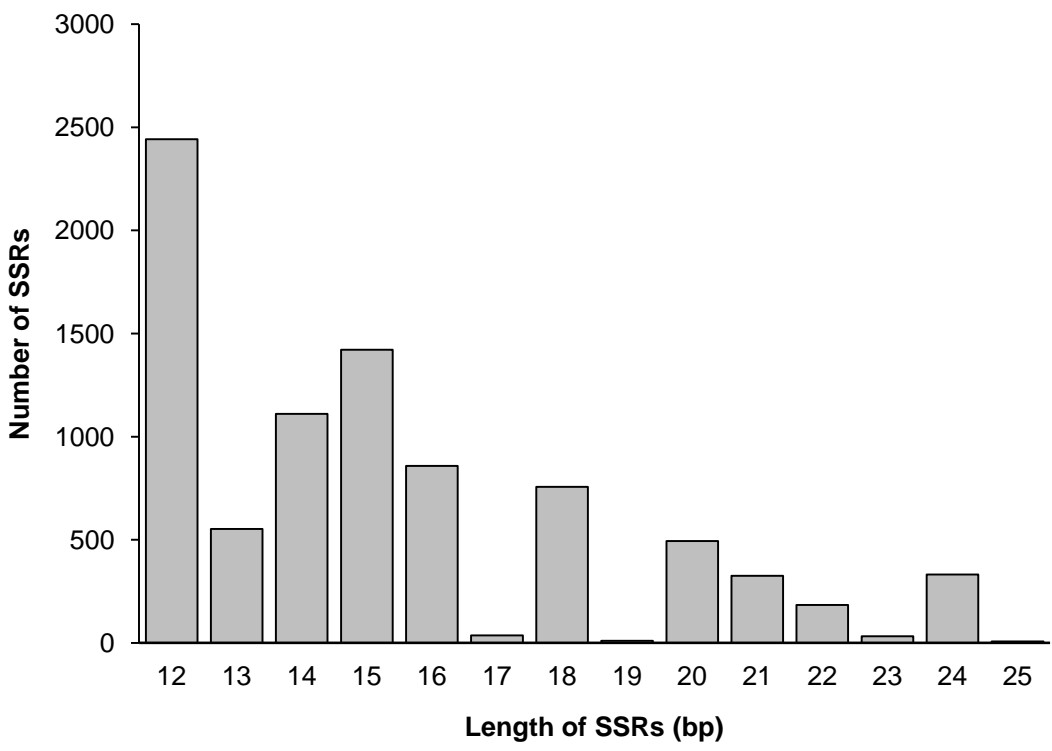

**Figure 6** The distribution of SSRs of different lengths.

sequences containing mononucleotides. Then, 74 primer pairs (dinucleotide, 20; tri-, 38; penta-, 3; hexa-, 13) were selected to validate the amplification across a composite sample of three individuals. A total of 60 primer pairs (81.08% of 74 primer pairs) showed stable and clear amplification. Meanwhile, the 14 remaining pairs with failed PCR produced multiple bands or showed unstable amplification. After polymorphism screening across 12 individuals, 23 primer pairs were found to be monomorphic and 37 were found to be polymorphic. Among 37 polymorphic primer pairs, 17 pairs of highly polymorphic and stable loci were selected for further polymorphism screening across 128 individuals from 11 populations, which covered the majority of habitats of these plants. For the 17 polymorphic loci, there were 2–6 alleles at each locus, with a total of 65 alleles. The $H_E$, $H_O$ and PIC per locus ranged from 0 to 0.196, 0 to 0.14 and 0.155 to 0.664, respectively. For the PIC values of the 17 polymorphic loci, eight pairs had highly informative scores (PIC > 0.50) and five pairs had weakly informative scores (0 < PIC < 0.25) (Table 2).

## Functional annotation of SSR-containing ESTs

The functional annotation distribution of SSR-containing ESTs centered on ESTs containing di- and tri-nucleotide SSRs (BC1–BC10). With the exception of one EST (BC12), the ESTs containing penta- and hexa- nucleotide SSRs (BC11 to BC17) had almost no functional annotations. However, the ESTs containing BC2, BC4 and BC12 were annotated by the KEGG analysis to vegetative dehydration/desiccation pathways (Table 3 and Table S1). Thus, these ESTs may contain the SSRs involved in regulating the function of

**Table 2** Characteristics of 17 polymorphic EST-SSR markers.

| Locus | Primer sequence 5'–3' | Repeat motif | $N_A$ | Size range (bp) | $H_E$ | $H_O$ | HWE[a] | PIC | GenBank accession no. |
|---|---|---|---|---|---|---|---|---|---|
| BC1 | F:GCAGTTCTGTGCAGTACCATACAT R:GGCTTCTGATCAGGTTTCTGAAT | $(TA)_6$ | 4 | 172–182 | 0.065 | 0.038 | 0.036* | 0.193 | Pr032805680 |
| BC2 | F:GAGATCCCAGATCCAGATCTTCT R:AACATTAATGGAAACACGTCGTC | $(TC)_6$ | 3 | 160–164 | 0.038 | 0.023 | 0.192 n.s | 0.423 | Pr032805689 |
| BC3 | F:ATTCGCTCTCTTGGTATGACTGT R:CCCAATTTGAAGTGTTGCTTTAC | $(TA)_6$ | 5 | 170–184 | 0.054 | 0.045 | 0.380n.s | 0.664 | Pr032805690 |
| BC4 | F:TATCAGCGTGTGTGAATAGTTGC R:TAACCTAAATTCGAATCCATCCA | $(TA)_7$ | 4 | 157–163 | 0.097 | 0.045 | 0.004** | 0.491 | Pr032805691 |
| BC5 | F:CAAACTTGGCTTAATACCATTCG R:CCATGATCATCTCTATTTCAGGC | $(TG)_9$ | 3 | 119–125 | 0.079 | 0.083 | 0.713 n.s | 0.469 | Pr032805692 |
| BC6 | F:CCTTAAGGAGATGCATTGTGAAT R:GTATGAAGGGCATCAACAATAGG | $(TC)_9$ | 3 | 159–169 | 0.000 | 0.000 | - n.c. | 0.299 | Pr032805693 |
| BC7 | F:GCTGAAAGTTGGTGATTGCTAGT R:AGTTATGTCTTCGCTTGCTTCAG | $(AT)_9$ | 4 | 166–178 | 0.120 | 0.125 | 0.087 n.s | 0.526 | Pr032805694 |
| BC8 | F:AACGTGAGAGTGCTAGTTCGGTA R:TCTTCCTCACTTTATCATCCACG | $(TGA)_5$ | 3 | 167–173 | 0.014 | 0.000 | 0.041* | 0.17 | Pr032805695 |
| BC9 | F:AGAAGAGGTACGACAGTTTGCTG R:TTCACGTCCGAATTCTTAGTCTC | $(GCG)_5$ | 2 | 156–159 | 0.059 | 0.064 | 1.000 n.s | 0.195 | Pr032805696 |
| BC10 | F:CACTGCACATAGAAGGAGGAGTT R:GTAATCGCCTACATGATTCATCC | $(GCG)_6$ | 5 | 108–129 | 0.081 | 0.076 | 0.146 n.s | 0.581 | Pr032805681 |
| BC11 | F:CAGCAGTATGTCGGGATTATTTC R:CCTCTGGTCATATTGCTGTTACC | $(TTTCT)_4$ | 2 | 123–133 | 0.000 | 0.000 | -n.c. | 0.155 | Pr032805682 |
| BC12 | F:AACAAGAGGGTCAGCTACAACAG R:CAGCAATGGTATTAGCAGAGGAC | $(CAGCAA)_4$ | 4 | 160–178 | 0.104 | 0.095 | 0.184 n.s | 0.549 | Pr032805683 |
| BC13 | F:ACCTTGACGATCCTTCATCTTCT R:TTATGTTCTCCATATCCGTCAGC | $(GGTGCG)_4$ | 6 | 132–174 | 0.124 | 0.095 | 0.161 n.s | 0.701 | Pr032805684 |
| BC14 | F:GGCAGCAATATAGCTCAAATACG R:ACCTGATCGTTCACAACTTCATC | $(GACAAG)_4$ | 4 | 170–188 | 0.196 | 0.083 | 0.000*** | 0.516 | Pr032805685 |
| BC15 | F:TCTTATTCAACACAACAGCCTGA R:GCTGCAGTTGATAATGAGAAGGA | $(ATGATA)_4$ | 5 | 151–175 | 0.157 | 0.140 | 0.228 n.s | 0.528 | Pr032805686 |
| BC16 | F:ACCAATGGTCTATATTTCAACGG R:GTGCCCCACATAGCTTCTATCTA | $(ATTACT)_4$ | 6 | 149–179 | 0.132 | 0.125 | 0.174 n.s | 0.643 | Pr032805687 |
| BC17 | F:TGACGAGGCTTCTACAGAATGAG R:ACAAACAACAAGATGGGAATCAT | $(CATCCT)_4$ | 2 | 137–143 | 0.034 | 0.045 | 1.000 n.s | 0.186 | Pr032805688 |

**Notes.**

$N_A$, number of alleles per locus across all populations; $H_E$, expected heterozygosity (mean value); $H_O$, observed heterozygosity (mean value); PIC, polymorphic information content; HWE, Hardy-Weinberg equilibrium.

[a]After Bonferroni correction, significant departures from Hardy–Weinberg equilibrium are indicated by *$P < 0.05$, **$P < 0.01$, ***$P < 0.001$. n.s., not significant; n.c., not calculated (*Clarke & Gorley, 2001*).

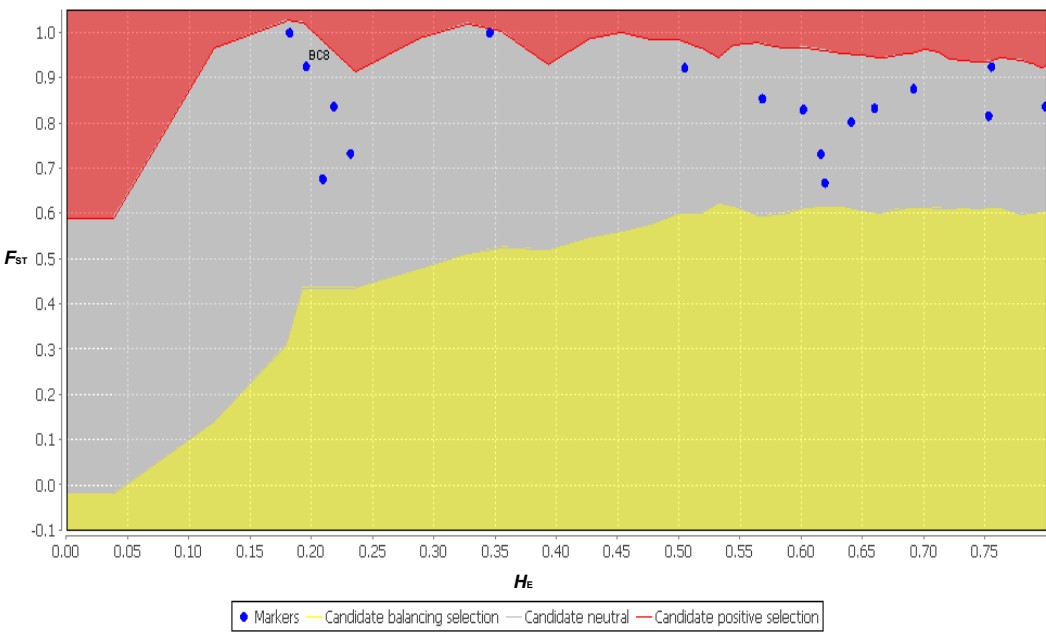

**Figure 7** The neutral test results for 17 primer pairs using $F_{ST}$ and $H_E$ from 11 populations using LOSITAN.

DT-related genes. Although the SSR variation of the functional markers did not agree with the neutral theory, the neutrality test conducted using LOSITAN showed that all 17 primer pairs agreed with the neutral theory (Fig. 7). These 17 primer pairs may have contained exactly neutral markers, or perhaps the sample size and randomness of sampling in this study was deficient. Therefore, increasing the sample size in future studies will provide us with more accurate results. It should be noted that compared with previous reports that identified EST-SSRs using NGS, the hexa- nucleotide SSR-containing ESTs, which comprised a relatively larger number of sequences with more polymorphic markers and fewer annotations, were unique to *B. clarkeana* (*Liu et al., 2013*; *Wang et al., 2013*; *Xiang et al., 2015*).

## DISCUSSION

### Gene expression characteristics and comparison of *B. clarkeana* with *B. hygrometrica*

This work is the first genetic study of *B. clarkeana* as a new plant resource of DT genes. Notably, a large amount of EST data were available, enabling a better understanding of gene expression in this species. *B. clarkeana* was compared with the related species *B. hygrometrica*, and both plants showed KEGG enrichment of vegetative dehydration/desiccation pathways; these results showed the common characteristics of metabolic pathways in DT plants. However, there were some differences between the transcriptome data of these two species. First, the GC content (45.43%) of *B. clarkeana* was higher than that of *B. hygrometrica*, which was close to the distribution centered value of coding sequences (*Matassi et al., 1989*). Likely due to the lack of dehydration stress, the
**Table 3 GO, COG and KEGG annotation of 17 SSR-containing ESTs.**

| Unigene[a] | Annotation | | | | |
|---|---|---|---|---|---|
| | GO[b] | | | COG | KEGG |
| | b. p. | c. c. | m. f. | | |
| BC1 | – | 6 | – | Function unknown | – |
| BC2 | 1 | – | – | – | Plant hormone signal transduction |
| BC3 | – | – | – | – | – |
| BC4 | 7 | – | 3 | General function prediction only | Plant hormone signal transduction |
| | | | | Signal transduction mechanisms | Plant-pathogen interaction |
| | | | | Transcription | |
| | | | | Replication, recombination, and repair | |
| BC5 | – | – | – | – | – |
| BC6 | – | – | – | General function prediction only | – |
| BC7 | 2 | 2 | – | General function prediction only | Biosynthesis of secondary metabolites |
| | | | | | Amino sugar and nucleotide sugar metabolism |
| BC8 | 12 | 4 | 4 | General function prediction only | – |
| | | | | Signal transduction mechanisms | |
| | | | | Transcription | |
| | | | | Replication, recombination and repair | |
| BC9 | – | – | 2 | Secondary metabolites biosynthesis, transport and catabolism | Biosynthesis of secondary metabolites |
| | | | | | Flavonoid biosynthesis |
| | | | | | Flavone and flavonol biosynthesis |
| | | | | | Sesquiterpenoid and triterpenoid biosynthesis |
| | | | | | Isoflavonoid biosynthesis |
| BC10 | 2 | 4 | 1 | Cell cycle control, cell division, chromosome partitioning | Metabolic pathways |
| | | | | | Endocytosis |
| | | | | | Ether lipid metabolism |
| BC11 | – | – | – | – | – |
| BC12 | 10 | 3 | 2 | General function prediction only | Plant hormone signal transduction |
| | | | | Posttranslational modification, protein turnover, chaperones | |
| BC13 | – | – | – | Transcription | – |
| BC14 | – | – | – | – | – |
| BC15 | 9 | 5 | 2 | – | – |
| BC16 | – | – | – | – | – |
| BC17 | – | – | – | – | – |

**Notes.**

b. p., biological_process; c. c., cellular_component; m. f., molecular_function.

[a]The name of each unigene is replaced with the name of the EST-SSR maker it contains.

[b]This table shows only the number of the GO terms in the ontology for unigenes. The details of the GO classification in three ontologies are shown in Table S1.

annotated unigene percentage (78.82% of all assembled unigenes) and the enrichment of GO (65.57% of all unigenes) and KEGG (49.12%) annotations in this study were much greater than those of *B. hygrometrica* (Nr, 47.09%; GO, 28.71%; KEGG, 24.43% (*Xiao et al., 2015*) and Nr, 66.6%; GO, 43.7%; KEGG, 15.1% (*Zhu et al., 2015*)). Second, due to different sequencing depths or selective gene expression at various physiological stages, the structural features of gene expression in *B. hygrometrica* was quite different under

various environmental pressures. *Zhu et al. (2015)* found that *B. hygrometrica* matched in the Nr database with *V. vinifera*, *R. communis* and *P. trichocarpa*, whereas *Xiao et al. (2015)* found shared genes in the genomes of *B. hygrometrica*, *Solanum tuberosum* and *Solanum lycopersicum* (Solanales). Thus, Solanales and *V. vinifera* could both contain the main components of the protein-coding genes of *B. hygrometrica*, which would be similar to the structural features of gene expression in *B. clarkeana* in the present study.

### EST-SSR characteristics of *B. clarkeana*

A significant number of SSRs were identified in the present study. A higher number of SSRs indicated stronger environmental adaptation capabilities (*Zalapa et al., 2012*); therefore, *B. clarkeana* should be highly adaptable to different environments due to the large number of SSRs contained in its ESTs. Moreover, the 'Transcription', 'Replication, recombination and repair' and 'Signal transduction mechanisms' reflect the strong ability of *B. clarkeana* to undergo environmental adaptation.

Intrinsic features (such as repeat number, motif size, and length) could influence the rate and probability of slippage. These features were the strongest predictors of microsatellite mutability (*Kelkar et al., 2008*). The increased probability of slippage and mutation rates may be due to, for example, a greater number of repeats (*Ellegren, 2004*; *Kelkar et al., 2008*), a greater length irrespective of the repeat numbers (*Webster, Smith & Ellegren, 2002*), and lengths that were inversely proportional to their motif sizes (*Chakraborty et al., 1997*). Additionally, the mutation rates might vary among SSRs with different motif compositions due to the dissimilarities of secondary DNA structures (*Baldi & Baisnee, 2000*). In this study, 37 pairs (dinucleotide, 13; tri-, 13; penta-, 2; hexa-, 9) of the 74 primer pairs (dinucleotide, 20; tri-, 38; penta-, 3; hexa-, 13) that were selected to validate the amplification results were polymorphic. The percentage of polymorphism was 65% in dinucleotides (13 of the 20 selected were polymorphic), 34.21% (13 of 38) in trinucleotides, 66.67% (2 of 3) in pentanucleotides and 69.23% (9 of 13) in hexanucleotides. As a result, in our study, SSRs with higher polymorphism rates were concentrated on shorter motifs with a higher number of repeats (dinucleotides, 65%) and longer motifs with fewer repeats (hexanucleotides, 69.23%; pentanucleotides, 66.67%). Our analysis confirmed that mutability might increase with both increased repeat number and greater length, as reported by *Baldi & Baisnee (2000)*.

Compared with other SSR and EST-SSR reports (*Choudhary et al., 2009*; *Li et al., 2012a*; *Li et al., 2012b*; *Yuan et al., 2012*; *Fu, Wang & Shen, 2013*), the observed number of polymorphic primers was actually higher, but the polymorphism level of the markers and the $H_O$, $H_E$, HWE and PIC values of the *B. clarkeana* population were still much lower in our study and were similar to those of *B. hygrometrica* (*Xiao et al., 2015*).These results could be attributed to two main reasons: first, the number of SSRs and polymorphisms of the DNA protein-coding sequence was expected to be lower than that in non-coding sequences, and the mutation rate within these regions was lower than that in other DNA sequences (*Blanca et al., 2011*; *Zalapa et al., 2012*). Second, *B. clarkeana* is a plant with a short stature that requires scattered light. As this plant grows on the north side of rock outcrops (mostly limestone) and in the shadow of trees and shrubs (*Chao et al., 2013*),

the long-distance dispersal potential of windborne seeds might be significantly reduced. Furthermore, the occurrence of biparental inbreeding could be universal in plants with high self-compatibility (*Li & Wang, 2005*), which would cause lower genetic variability within populations of *B. clarkeana*.

## CONCLUSIONS

In this study, 91,449 unigenes were detected by NGS transcriptomics. A total of 8,563 SSRs were identified from 7,610 unigenes, 72,087 unigenes were successfully annotated to protein databases, and polymorphic primer pairs of EST-SSRs were also developed. These results indicated that transcriptome sequencing is a highly efficient method of EST-SSR identification in non-model species that lack a reference genome and associations with functional genes. Therefore, by characterizing phenotypic features, these species can be identified (*Li et al., 2002*). These data will accelerate our identification of functional genes and genetic variation in DT plants, including *B. clarkeana*. In addition, polymorphic primer pairs can continue to be developed from the remaining primers of EST-SSRs. The large-scale transcriptome dataset is a powerful resource for functional gene marker-assisted selection and DT exploration in *Boea* plants.

## ACKNOWLEDGEMENTS

We are grateful to Cunhai Li and Fei Tan (Guanshan National Nature Reserve, JiangXi, China) for their assistance with the sampling.

### Funding

This work was supported by the Cultivation Foundation for Young Talents at Basic Research of Anhui Normal University (No. 2015rcpy35, awarded to Ying Wang) and the National Natural Science Foundation of China (No. 31400291, awarded to Kun Liu). The funders had no role in study design, data collection and analysis, decision to publish, or preparation of the manuscript.

### Grant Disclosures

The following grant information was disclosed by the authors:
Cultivation Foundation for Young Talents at Basic Research of Anhui Normal University: 2015rcpy35.
National Natural Science Foundation of China: 31400291.

### Competing Interests

The authors declare there are no competing interests.

### Author Contributions

- Ying Wang conceived and designed the experiments, performed the experiments, analyzed the data, contributed reagents/materials/analysis tools, wrote the paper, prepared figures and/or tables, reviewed drafts of the paper.

- Kun Liu and De Bi contributed reagents/materials/analysis tools.
- Shoubiao Zhou conceived and designed the experiments, reviewed drafts of the paper.
- Jianwen Shao conceived and designed the experiments, analyzed the data, contributed reagents/materials/analysis tools, reviewed drafts of the paper.

## DNA Deposition

The following information was supplied regarding the deposition of DNA sequences:

The raw reads squences and primer pairs described here are accessible via GenBank accession numbers SRX1600046 and Pr032805680–Pr032805696.

## Data Availability

Wang, Ying (2017): Datas of the transcriptome and EST-SSR development in *Boea clarkeana*. figshare.

https://doi.org/10.6084/m9.figshare.4197987.v1.

## Supplemental Information

Supplemental information for this article can be found online at http://dx.doi.org/10.7717/peerj.3422#supplemental-information.

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
