# Peer review of "Characterization of the transcriptome and EST-SSR development in Boea clarkeana, a desiccation-tolerant plant endemic to China"

_PeerJ, doi:10.7717/peerj.3422_

## Round 0.1 · original submission · Major Revisions

Please, provide a revised version of the manuscript according to the suggestions of the reviewers. Please, explain all changes in the manuscript in a rebuttal letter.

Reviewer 1 ·

Basic reporting

No comment

Experimental design

No comment

Validity of the findings

No comment

Additional comments

The article entitled “Characterization of the transcriptome and EST-SSR development in Boea clarkeana, a desiccation-tolerant plant endemic to China” reported the transcriptome characterization of Boea clarkeana and EST-SSR identification in this plant. The data generated from Illumina sequencing and the data analyzed soundly. Although this manuscript could be considered for publication, there are still several suggestions for the revision.
1. The title is “Characterization of the transcriptome and EST-SSR development in Boea clarkeana, a desiccation-tolerant plant endemic to China”. According to the result and discussion, the content related to the desiccation-tolerant features of this plant was so limited. The data associated with this aspect need to be analyzed or discussed.
2. There were several spell errors in the manuscript. Such as the word of MgCl2 in Line 177 should be MgCl2. Please check out these spell errors carefully throughout the manuscript.
3. The sentence of “Based on Nr annotations, 59,962 unigenes (65.57% of all unigenes) were assigned to three ontologies and subdivided into 55 subcategories with 501,897 functional GO terms of GO classifications (Fig. 3)” between the Line 224 to 226 need to rewrite. This meaning of this sentence is complex. First, if the GO analysis method was based on the Nr annotations? Second, the 501,897 functional GO terms, why so many terms?
4. Most of the functional annotation are so commonly. The date could be analyzed in detail. Such as, what are the highly expressed genes in the leave of B.clarkeana? If these genes are related to the desiccation-tolerant of this plant?
5. The EST-SSRs were identified from the unigenes of B.clarkeana. The predicted function of these unigenes containing SSRs could be depicted and discussed, which will provide the possible application for these SSR.

Reviewer 2 ·

Basic reporting

The authors report transcriptome data for the desiccation tolerant plant Boea clarkeana. The data are useful for the scientific community working on desiccation tolerance, as they broaden the database for desiccation tolerant organisms.
The authors also develop SSR markers for screening plant populations. However, the markers were not tested on a population, this then questions how useful it is to publish the markers in this manuscript. these markers

Experimental design

The generation and analysis of the transcriptome data corresponds to stae of the art experimental procedures.
The data deposition needs to be controlled, which is very important. The authors write that the data will be released in March 2018, this is not acceptable, the data need to be accessible at the time when this manuscript is published.

Validity of the findings

The authors do not provide a comparison to published transcriptome and genome data from Boea hygroscopica, this should be included and will make the manuscript much stronger.
The authors mention functional studies several times, however no functional studies are provided, the authors provide a description of the transcriptome

Additional comments

The following points should be included in a revision:
-include a comparison with published transcriptome data from Boea hygroscopica
-demonstare usefulness of selected SSR markers
-make transcriptome data available at the time of publication
-there are a number of linguistic errors throughout the manuscript which should be corrected, also the references should be controlled, some are not cited at the correct place.

Reviewer 3 ·

Basic reporting

The authors describe the characterization of the transcriptome of a desiccation-tolerant plant, Boea clarkeana. This work is quite reminiscent of the work already presented for Boea hygrometrica, and the authors make little effort in emphasizing what aspects are unique to B. clarkeana versus B. hygrometrica.

The data generated is of course valuable to the B. clarkeana research community, and the SSR and SNPs identified helpful for marker development. However, simply reporting the original data or very preliminary analytical data, as the case in this study, made the current study very superficial.

The English should be improved by somebody proficient. I was a little confused by some of the long sentences or inappropriate use of words, e.g. lines 330-332, 335, 359, 363, 393-397. In addition, I feel very sorry to see the first sentence of Abstract, which might be copied from the paper of Xiao et al., Proc Natl Acad Sci USA 2015, 112(18): 5833-5837.

Experimental design

The materials for RNA-seq is young leaves of three individual B. clarkeana plants without any desiccation-tolerance-related treatments. The main reason for sequencing B. clarkeana is its high desiccation tolerance (DT). If no DT treatments, how to find DT-related genes? Hence, I don’t think this study could improve our understanding of the genetic basis of DT in B. clarkeana.

Validity of the findings

The paper very descriptive. It is not so enlightening and showed no exciting findings about the transcriptome and the SSRs.

Additional comments

1. The introduction needs more detail. The authors should provide more justification for their study.
2. The authors also developed EST-SSR molecular markers. Of course, they can do this. However, the authors didn’t explain why they do this job on B. clarkeana.
3. Similarly, they also detected SNPs and said nothing about the importance and necessity of SNP calling in the MS.

---

## Round 0.2 · accepted · Accept

The revised version of the manuscript has improved and the authors addressed all concerns of the reviewers.